# Immune Response and Breakthrough Infection Risk After SARS-CoV-2 Vaccines in Patients with Hemoglobinopathy: A Single Center Experience

**DOI:** 10.3390/vaccines13020111

**Published:** 2025-01-23

**Authors:** Andrea Duminuco, Anna Bulla, Rosamaria Rosso, Maria Anna Romeo, Daniela Cambria, Enrico La Spina, Benedetta Ximenes, Cesarina Giallongo, Daniele Tibullo, Alessandra Romano, Francesco Di Raimondo, Giuseppe A. Palumbo

**Affiliations:** 1Hematology Unit with BMT, A.O.U. Policlinico “G. Rodolico—San Marco”, Via S. Sofia 78, 95123 Catania, Italy; cambriad@tiscali.it (D.C.); enricolaspina@outlook.it (E.L.S.); sandrina.romano@gmail.com (A.R.); francesco.diraimondo@unict.it (F.D.R.); 2Thalassemia Unit, A.O.U. Policlinico “G. Rodolico—San Marco”, Via S. Sofia 78, 95123 Catania, Italy; anna.bulla24@gmail.com (A.B.); rosellinarosso@gmail.com (R.R.); m.romeo@ao-ve.it (M.A.R.); beximene@tin.it (B.X.); palumbo.gam@gmail.com (G.A.P.); 3Dipartimento di Scienze Mediche, Chirurgiche e Tecnologie Avanzate “G.F. Ingrassia”, University of Catania, 95123 Catania, Italy; cesarina.giallongo@unict.it; 4Dipartimento di Scienze Biomediche e Biotecnologiche, University of Catania, 95123 Catania, Italy; d.tibullo@unict.it; 5Dipartimento di Specialità Medico-Chirurgiche, CHIRMED, Sezione di Ematologia, University of Catania, 95123 Catania, Italy

**Keywords:** COVID-19, SARS-CoV-2, mRNA vaccine, hemoglobinopathy, thalassemia, sickle cell disease, immune response, breakthrough infections

## Abstract

Background: Immune system impairment is frequently reported in patients affected by hemoglobinopathies due to various mechanisms, including iron accumulation, antigenic stimulation due to numerous transfusions, chronic hemolysis, and a general hyperinflammatory state. For these reasons, the antigenic immune response after a vaccine risks being ineffective. Methods: We evaluated the anti-spike IgG production after two doses of vaccine for SARS-CoV-2 in patients affected by hemoglobinopathies. Results: All 114 enrolled patients (100%) developed adequate antibody production, with a median value of serum IgG of 2184.4 BAU/mL (IQR 1127.4–3502.9). The amount of antibody was unrelated to any other clinical characteristics evaluated, including transfusion dependence or non-transfusion dependence, age, gender, disease type, ferritin, blood count, spleen status, and therapy with hydroxyurea or iron chelators (in all the cases *p* > 0.05). Moreover, 47 (41.2%) patients developed breakthrough SARS-CoV-2 infection during the first 2 years of follow-up after vaccination, all with a mildly symptomatic course, without requiring hospitalization or experiencing a significative drop in hemoglobin values, allowing for a slight delay in their transfusion regimen. Conclusion: Vaccination against COVID-19 is safe and effective for patients affected by hemoglobinopathies, ensuring adequate protection from severe infection.

## 1. Introduction

Hemoglobinopathies are a heterogeneous group of inherited disorders characterized by alterations in hemoglobin synthesis, including β-thalassemia and sickle cell disease (SCD—thalasso-drepanocytosis and homozygous sickle cell anemia) as the most common conditions. The clinical course is variable and characterized by different grades of anemia from birth, followed by various complications potentially affecting every organ system.

In the last decades, therapeutic advances have greatly improved patients’ survival rates and quality of life [1,2]. Recently, interest has been redirected to understanding other pathological aspects of these disorders, such as patients’ immune system status, which, in past decades, has yet to be fully investigated. It is well described that thalassemia major patients are at increased risk of infections. Underlying pathophysiological mechanisms of the disease, such as ineffective erythropoiesis (IE), chronic hemolysis and anemia, multiple blood transfusions, deferoxamine therapy, iron overload, and tissue hypoxia, contribute to increased susceptibility to viral and bacterial infections, increasing the morbidity and mortality in this setting of patients [3]. Furthermore, multiple red blood cell transfusions, the most common therapeutic approach in these diseases, and iron overload are known to be associated with immune suppression and systemic inflammation, in addition to the well-known transfusion-related complications [4]. On top of that, most patients (above all in the past) have undergone splenectomy, and others suffer from functional asplenia: these two conditions impact negatively on the immune response.

During the recent SARS-CoV-2 pandemic, several studies were performed to determine the mortality and incidence rates of SARS-CoV-2 infection among patients with hemoglobinopathies. Patients with pre-existing comorbidities are at greater risk for more aggressive disease than healthy people, and hemoglobinopathies are no exception [5]. In December 2020, COVID-19 mRNA vaccines were approved for use, and all patients with hemoglobin disorders were included among the “vulnerable” groups to be prioritized for the vaccination. Concerning response against COVID-19 vaccination, the first evidence was reported by Radhwi et al., who described the presence of neutralizing antibodies (nAb) in 84.5%, without correlation with age, gender, vaccine type, spleen status, use of hydroxyurea as treatment, and high value of ferritin [6].

In our study, we investigated if this setting of patients could achieve a protective antibody level against SARS-CoV-2 after vaccination in a large series of patients followed for their hemoglobinopathy in a single center, reporting subsequent risk of incurring breakthrough infection and evaluating the need for changes to the transfusion regimen.

## 2. Materials and Methods

### 2.1. Aim of the Study

Real-life data on immunization with COVID-19 vaccines in patients with thalassemia and sickle cell disease are few in number. This study aims to describe our center’s experience assessing the level of protective antibodies against SARS-CoV-2 developed by patients after vaccination and evaluating the outcome of breakthrough infection (after the vaccination). In March 2021, in Italy, patients with hemoglobinopathies were included among the vulnerable category and promptly received vaccination with the available COVID-19 virus mRNA vaccine (Comirnaty, Pfizer–BioNTech, Cambridge, MA, USA). The data were collected until March 2023.

### 2.2. Patients

In this observational single-center study, we enrolled 114 adult patients with hemoglobinopathies. In total, 98 (86%) patients were affected by β-thalassemia (63 thalassemia major and 35 thalassemia intermedia, 88 of which were transfusion-dependent), 10 by combined forms of thalassemia (such as β-thalassemia/HbE and α-thalassemia), 2 homozygous sickle cell anemia, and 14 thalasso-drepanocytosis. The cohort’s median age was 46 years (range of 22–82), with a balanced sex distribution (62 males and 52 females), suggesting a representative sample of adult patients with hemoglobinopathies in the studied population.

Sixty-seven patients had undergone surgical splenectomy before the vaccine administration due to their hematological condition. Most patients (98 of 114) receive transfusions with filtered red blood cell concentrates (RBCs) at regular intervals (mean 37 units of RBC transfused/year). Four patients were undergoing hydroxyurea therapy. A double dose, 21 days apart, of mRNA Comirnaty vaccine (30 μg per dose) was administered to 108 of 114 patients enrolled. Six patients received a single dose of the vaccine because of the previous SARS-CoV-2 infection, confirmed by the assessment of IgG-positivity before the vaccination.

Other baseline data are reported in Table 1.

Patients with recent SARS-CoV-2 infection or high immunosuppressive treatment (≤28 days), history of chronic inflammatory conditions, autoimmune disorders, and immunodeficiency beyond hemoglobinopathy were excluded.

After administration, patients were observed for 1 h to detect any possible immediate reactions. Immunity from infection was expected one week after the second dose of the vaccine.

As mentioned earlier, 6 patients experienced SARS-CoV-2 (based on nucleic acid testing of pharyngeal swab samples) before the vaccination. All had close contact with an infected family member or community exposure. Among them, 5 reported mild symptoms during quarantine and clinical monitoring. Fever, cough, headache, fatigue, gastrointestinal symptoms, tachypnea/dyspnea, anosmia/hyposmia, and myalgia were present. Most of the patients were not hospitalized. One patient reported severe pneumonia requiring oxygen therapy and hospitalization. Negativity was assessed with a molecular swab. The median follow-up after vaccination was 23.7 months.

### 2.3. Biologic Samples and Laboratory Methods

After obtaining informed consent from each patient, whole blood and sera samples (2–3 mL) from peripheral blood were collected during pre-transfusion routine laboratory tests to detect antibody levels of the COVID-19 virus. Blood samples were obtained when subjects were considered fully vaccinated, at least 15 days after receiving the second dose of vaccine, and no longer than 30 days. Sera were immediately frozen at −20 °C until analysis.

The SARS-CoV-2 negativity at every time point was assessed through molecular swab, analyzed according to the worldwide used sampling approaches and techniques [7,8,9,10,11,12].

Anti-COVID-19 virus antibody concentrations were determined by measuring the serum IgG neutralizing Ab levels against the RBD portion of the virus spike protein (anti-RBD), based on the IgG II Quant kit (Abbott, Chicago, IL, USA). In this antibody CMIA test, paramagnetic microparticles coated with SARS-CoV-2 antigens bind to IgG antibodies targeting the virus’s spike protein in human serum or plasma samples. Upon adding an acridinium-labeled anti-human IgG (mouse, monoclonal) conjugate, the resulting chemiluminescence, measured in relative light units (RLU), is compared to an IgG II calibrator/standard. This comparison determines the response intensity, corresponding to the IgGSP quantity present [13].

Before administration of the first vaccine dose, IgG against COVID-19 nucleocapsid proteins (anti-N) were researched to rule out a subject with prior or ongoing virus immunization before vaccination, based on the assumption that vaccinated patients are anti-RBD positive and at the same time anti-N negative. On the other side, patients infected by the virus should be anti-RBD and anti-N positive [14].

A measurement finding above 7 binding antibody units (BAU) was considered a cutoff for seropositive interpretation, according to the literature data reported [15,16]. The conversion to BAU for Abbott was BAU/mL = 0.142 Abbott value in AU/mL. Vaccine efficacy, in terms of protective immunity, is correlated to the presence of neutralizing antibodies [17,18,19]. The diagnosis of SARS-CoV-2 infection was defined by positive polymerase chain reaction or rapid antigen swab testing following local prevention guidelines.

### 2.4. Statistical Analysis

The qualitative results were summarized and presented as counts and percentages to provide a clear overview of the categorical data. Descriptive statistics were employed to analyze and interpret the findings, ensuring an accurate representation of the underlying patterns and trends in the data. Statistical significance was determined using a *p*-value threshold of less than 0.05, which was considered indicative of meaningful differences or associations within the dataset.

For continuous variables, the results were expressed as the median, along with the interquartile range, considering the non-normally distributed data. The calculations and statistical analyses were conducted using the R-commander software 4.3.3 (R Foundation for Statistical Computing, Vienna, Austria, available at https://www.R-project.org/).

## 3. Results

### 3.1. Vaccine Antibodies’ Response

The vaccine was well tolerated, with only minor side effects (fever, pain at the injection site, or local redness or swelling) that resolved spontaneously in most patients. No symptomatic infection was observed during the study period.

In the basal cohort, one patient (with a previous SARS-CoV-2 infection) affected by thalassemia major resulted in being positive for IgG against COVID-19 nucleocapsid proteins, while the remaining 113 were negative. The median anti-N IgG was 0.04 [range, 0.01–1.69], as reported in Appendix A.

At the end of the two-dose standard vaccination cycle, the median value of serum anti-RBD IgG was 2184.4 BAU/mL [IQR, 1127.4–3502.9]. All patients developed an IgG-neutralizing antibody level against the RBD portion of the spike protein superior to 7 BAU/mL (Figure 1).

The minimum measured value was 27.0 BAUI/mL. Only nine patients reported a <500.0 BAUI/mL value (four thalassemia intermedia, three thalassemia major, and two thalasso-drepanocytosis). The patient with previous anti-N positivity showed a median IgG value of 2834.7 BAU/mL (29% above the overall median).

Raw data are reported in Appendix A.

Despite evidence suggesting that patients with hemoglobinopathies may experience an impaired immune response, all study participants demonstrated an adequate antibody response to the mRNA COVID-19 vaccine. This robust response was consistent across various patient subgroups, with no significant differences observed based on sex, age, type of hemoglobinopathy, whether the patient was transfusion-dependent, prior splenectomy status, iron chelation therapy, or other concurrent treatments (*p* > 0.05).

Additionally, biochemical analysis revealed no significant relationship between ferritin or transferrin levels and the antibody response (*p* = 0.432). This suggests that these biochemical markers were not predictive of the vaccine-induced antibody response in this cohort. A detailed summary of the correlation between patients’ baseline characteristics and BAU is presented in Table 1, highlighting the consistency of the immune response across diverse clinical and demographic variables.

### 3.2. Breakthrough Infection

The incidence of breakthrough infection after vaccination was reported in 47 (41.2%) patients. Compared to the vaccine date, the median time to onset was 319 [13–656] days. The median duration of positivity was 13 [9–21] days. No patient reported severe complications or required hospitalization for life-threatening therapies. Concerning the hemoglobin value at the infection and the nadir during the positivity, no statistically significant reduction was reported (9.8 [9–11.1] versus 9.6 [8.6–10.9] g/dL), as evidenced in Figure 2. According to our policy, only two patients were transfused for anemia and profound asthenia related to basal hemoglobinopathy and concurrent fever due to the infection.

No differences were found comparing the patients who experienced a breakthrough infection with those who did not, dividing patients according to BAU value, history of splenectomy, type of hemoglobinopathies, and transfusion need status (all with *p* > 0.3), suggesting that the SARS-CoV-2 infection can happen beyond the clinical history, and the vaccine can offer adequate protection.

## 4. Discussion

Infections are one of the leading causes of morbidity and mortality in patients with hemoglobinopathies, described in β-thalassemia as the second most common cause of death, preceded by cardiac failure and hepatic disease [20]. The immune system status of these patients has not always been thoroughly investigated. Still, undoubtedly, the well-known increased susceptibility of patients with thalassemia to infections reveals an underlying immune system dysfunction. Indeed, several immune abnormalities have been described in β-thalassemic patients, both quantitative and functional, involving various components of the innate and adaptive immune response. Specifically, the T-lymphocyte compartment is characterized by an increased number and activity of suppressor T cells (CD8) with a concomitant reduction in proliferative activity of helper T cells (CD4), presenting a decreased CD4/CD8 ratio.

Moreover, a defective Natural Killer (NK) cell function is reported, with more B-lymphocytes with increased activity but reduced differentiation. These patients suffer from impaired immunoglobulin secretion with high levels of IgG, IgM, and IgA; abnormalities in chemotaxis and defective phagocytosis in neutrophils and macrophages and, finally, suppressed activities of the complement system, with reduced levels of C3 and C4. All of these anomalies have been attributed to the disease and therapeutic interventions [21,22]. Iron overload due to the physiopathology of the disease itself and the frequent and chronic need for blood transfusions has been implicated as the principal immunodeficiency factor in β-thalassemia. Free iron leads to numerous complications, such as heart failure, liver damage, and endocrine abnormalities, interfering with the immune balance and favoring the survival of infectious organisms [23]. Interestingly, immune system abnormalities that have been described in conditions characterized by iron overload as hemochromatosis include decreased phagocytosis by the monocyte–macrophage system; alterations in T-lymphocyte subsets, with an enhancement of CD8 and suppression of CD4; impairment of immunoglobulin secretion; and suppression of complement system function [24,25,26].

Moreover, hypertransfusion regimens applied to thalassemia major patients lead to continuous allo-antigenic stimulation, significantly impairing the immune balance [27,28]. Besides direct exposure to the risk of transfusion-transmitted infections, multiple transfusions have been associated with autoimmune hemolysis, T- and B-lymphocyte alterations, disruption in the pattern of cytokine production, and modification of monocyte–macrophage functions [29].

On the other side, splenectomy (currently reserved for patients with marked symptoms related to hypersplenism) and hypo-functional spleen as a disease complication play a significant role in immune system modifications (with a significantly high absolute lymphocytic count and low level of IgM memory B cells), leading to an increased risk to complicated infections in thalassemia (even if it was not reported in our cohort, evidencing the efficacy of SARS-CoV-2 vaccination) [30,31,32,33,34].

In March 2020, the World Health Organization (WHO) declared the pandemic outbreak caused by SARS-CoV-2, with elderly and debilitated subjects identified as patients at higher risk. Hemoglobinopathy patients are more susceptible to virus infection than the general population, with an increased risk of a severe course, given the immunocompromised state and the many comorbidities they usually present [5].

In SCD, increased vulnerability to COVID-19 infection is likely due to compromised immunity resulting from impaired spleen function, systemic vasculopathy, hypercoagulability, and elevated risk of thrombosis, and patients with thalassemia often present multi-organ damage because of their immune dysfunction, iron overload, chronic anemia, and hypoxemia [35]. An important role can also be performed by altered tissue oxygen transport, iron metabolism, and elevated levels of oxidative stress [36,37]. Another relevant aspect of COVID-19 infection in hemoglobinopathies, especially SCD and thalassemia, is that patients are usually treated with hydroxyurea, a cytotoxic agent with possible immunocompromising effects, potentially worsening the outcome of the COVID-19 disease in these patients [38].

Despite these data, an early study of a small cohort of COVID-19-positive patients with thalassemia in Northern Italy showed relatively mild-to-moderate clinical courses when compared to the general population, with all infected thalassemia patients cured [39], in contrast to a study observed in Iran on a similar series with different severity and mortality [40]. In fact, differences in SARS-CoV-2 immunity among patients with hemoglobinopathies likely stem from regional factors such as healthcare access, prevalent viral variants, and vaccination rates. These patients may receive regular monitoring and vaccination in well-resourced areas, supporting better immune responses. In contrast, limited healthcare in some regions leads to unmanaged complications, weakening immunity. The regional dominance of specific viral variants also affects immune responses, as does varying vaccine access and types used. Environmental factors and chronic inflammation specific to hemoglobinopathies also impact immunity, making outcomes diverse worldwide.

In December 2020, the advent of RNA vaccines triggering an immune response by delivering a protein that produces antibodies to the COVID-19 virus completely changed our approach to this type of infection and the outcome of patients, especially in the setting of immunocompromised hematological subjects, where the treatments were inefficacy. Evidence to date has shown that the effectiveness of the two vaccines in preventing infection by the COVID-19 virus ranges from 94% to 95% [41]. In the setting of hematological patients, we previously studied and reported the risk of vaccine exacerbating an underlying coagulative disorder (i.e., hemophilia or thrombotic thrombocytopenic purpura [42,43]), and the role of the disrupted immune system and the previous or concomitant therapy in not allowing an adequate antibody response to the administration of the vaccine is known [44,45,46,47,48].

In the general population, the degree to which the humoral immune response contributes to vaccine efficacy remains unclear [49]. The involvement of other components of the immune system, particularly cellular immunity mediated by CD4+ and CD8+ T cells, which are known to be activated by vaccination, should also be considered. This aspect, however, was beyond the scope of our study. CD4+ T cells are pivotal in coordinating and integrating immune responses, while CD8+ T cells play a critical role in eliminating virus-infected cells [50]. Importantly, both memory and cytotoxic T-cells specific to viruses have been shown to persist for more than 15 years, providing long-lasting protection to immunized individuals [51].

The COVID-19 vaccine can induce both humoral and cellular immune responses, even if a robust and specific T-cell response is more likely to occur in patients who also generate a broad and functional humoral immune response [52,53].

Moreover, during the SARS-CoV-2 emergency, our management strategy included supportive transfusion therapy even during the patient’s positivity period. For this purpose, an area with suitable characteristics for the isolation and safety of the patient has been identified, with medical and nursing staff equipped with adequate protection.

However, we observed that, except for eight cases with a reduction > 0.5 g/dL, the patient’s Hb values did not significantly reduce compared to the basic during the positivity days. So, subsequently, we decided to postpone the transfusion of blood components until the patient becomes negative if the hemoglobin at the blood count check, always performed in dedicated areas and safety for the patient and staff, showed values ≥ 9.5 g/dL.

Our study confirms these data in a relatively large series of patients, showing that the underlying immunological alterations in patients affected by hemoglobinopathies do not significantly impair response to SARS-CoV-2 vaccination, preferring this prophylaxis in place of different strategies, such as tixagevimab–cilgavimab [54,55], antivirals [56,57], or other immunosuppressors (such as JAK inhibitors) [58,59,60].

Recently, Ghoti et al. reported significantly higher antibody levels in β-thalassemia patients compared to controls both post-infection (1299.0 vs. 555.7 AU/mL, *p* = 0.009) and post-vaccination (8404.0 vs. 2785.6 AU/mL, *p* = 0.015). Regarding breakthrough infections, they reported 61.9% of infections during the same follow-up, slightly higher than our cohort [61]. Also, these findings highlight the consistent immunogenicity of mRNA vaccines.

Considering the criticism of this study, of the patients included, only two were affected by SCD, while the majority had thalassemia. This distribution likely reflects the ethnic composition of the cohort, as thalassemia is more prevalent in populations of Mediterranean, Middle Eastern, and Southeast Asian origin, whereas sickle cell disease is more common in individuals of African descent. No differences in our cohort were reported between foreign and Italian residents, who were vaccinated at a rate similar to previously reported national data [62].

Aware of the limitations, such as the absence of a control group, the small number of sub-cohorts (such as the 2 SCD patients) that does not allow for specific sub-analysis, associated with the consideration that some samples exceeded the assay’s calibration curve (>40,000 AU/mL or >5680 BAU/mL—it may have influenced the accuracy of central tendency measures and error metrics), we report the efficacy of vaccination also in this setting of patients, avoiding any drop in hemoglobin value in case of breakthrough infections, contrary to the data reported before the vaccine era, allowing the transfusion regimen to be slightly delayed for each patient without significant risks [63].

## 5. Conclusions

Vaccination against SARS-CoV-2 has shown to be safe and immunogenic in thalassemia major and SCD patients. Other strategies, such as monoclonal antibodies and antivirals, should be used in cases of breakthrough infections.

## Figures and Tables

**Figure 1 vaccines-13-00111-f001:**
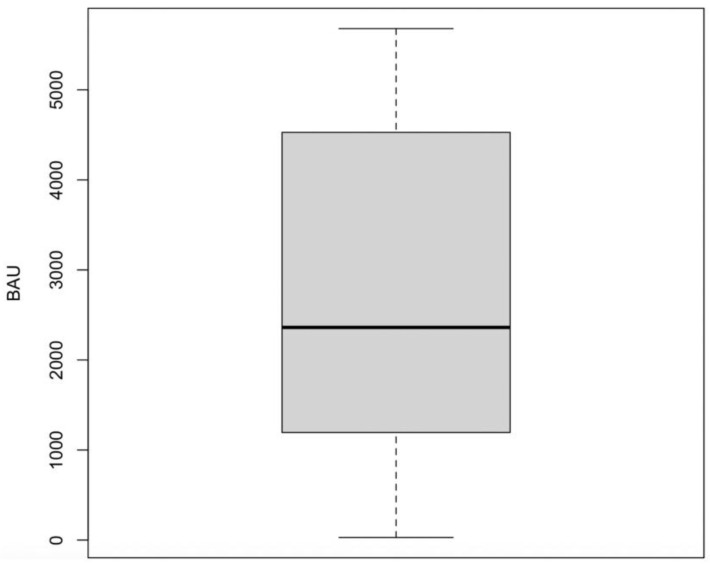
Boxplot of spike IgG antibodies in correlation after two doses of vaccine BNT162b2 (Pfizer-BioNTech) COVID-19 vaccine. BAU: binding antibody units.

**Figure 2 vaccines-13-00111-f002:**
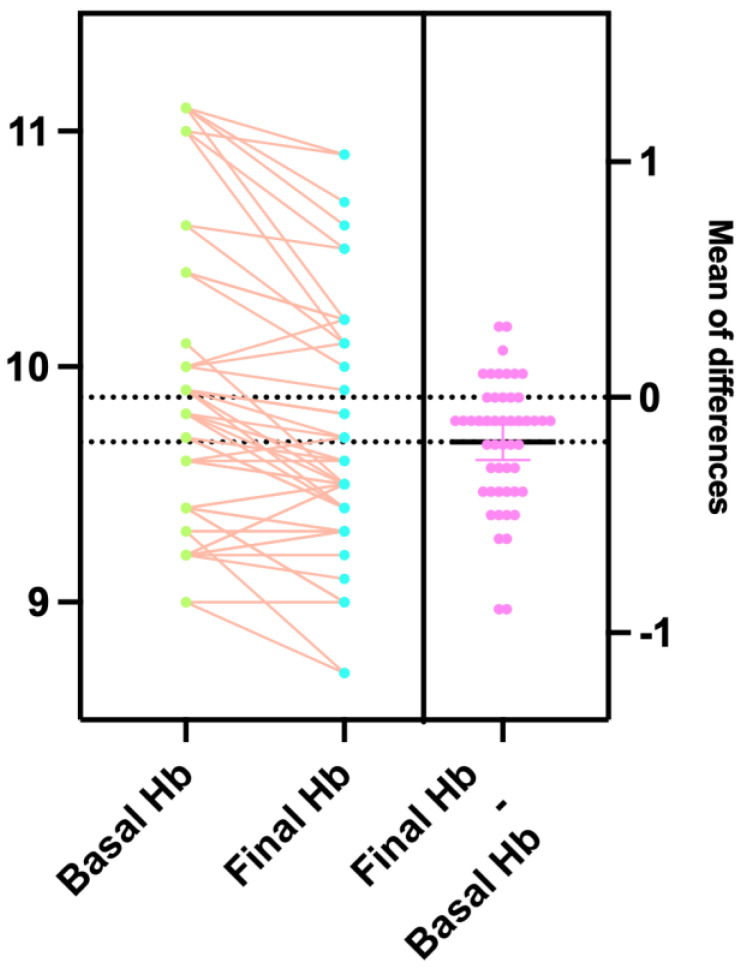
Fluctuation of hemoglobin value measured in 47 patients who experienced breakthrough infection. Hb: hemoglobin. In green were reported the basal values and in light blue the final. Every pink point represents a patient.

**Table 1 vaccines-13-00111-t001:** Baseline characteristics of the 114 enrolled patients and the demonstration of the absence of correlation between BAU value and variables through correlation matrix. RBCs, red blood cells; HU, hydroxyurea; IQR, interquartile range.

	At 1st Vaccine Dose *n* = 114 (Median) [Range]	*p*-Value
**Median age, years (range)**	46 (22–82)	0.2281
**Sex M/F, *n* (%)**	62 (54.4)/52 (45.6)	0.6465
**Diagnosis**		0.3383
**β-Thalassemia, *n* (%)**	98 (86.0)	
**Thalassemia major, *n* (%)**	63 (55.3)	
**Thalassemia intermedia, *n* (%)**	35 (30.7)	
**Combined thalassemia, *n* (%)**	10 (8.8)	
**Sickle cell anemia, *n* (%)**	2 (1.7)	
**Thalasso-drepanocytosis, *n* (%)**	14 (12.3)	
**RBC transfusion-dependent, *n* (%)**	98 (86.0)	0.7601
**RBC transfusion-independent, *n* (%)**	16 (14.0)	0.663
**HU ongoing treatment, *n* (%)**	4 (3.4)	0.6205
**Prior splenectomy, *n* (%)**	67 (58.7)	0.228
**Iron-chelation ongoing therapy, *n* (%)**	89 (78.1)	0.1379
**Median ferritin value, ng/mL [IQR]**	308 [153–662]	0.4322
**Median transferrin value, mg/dL [IQR]**	267 [220–315]	0.504
**Median WBC value, 10^3^/mmc [IQR]**	10.02 [6.93–14.25]	0.905
**Neutrophils, 10^3^/mmc [IQR]**	5.60 [4.06–8.33]	0.7979
**Lymphocytes, 10^3^/mmc [IQR]**	2.80 [1.98–4.59]	0.1507
**Median PLT value, 10^3^/mmc [IQR]**	429 [268–581]	0.0748

## Data Availability

The data presented in this study are available upon request from the corresponding author. The data are not publicly available due to privacy and ethical restrictions.

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
