# Peer review of "Immune Response and Breakthrough Infection Risk After SARS-CoV-2 Vaccines in Patients with Hemoglobinopathy: A Single Center Experience"

_vaccines, 2025, doi:10.3390/vaccines13020111_

Round 1

Reviewer 1 Report

Comments and Suggestions for Authors

Dear authors, 

your research is important, clearly designed and presented. I have only minor comments.

1. Figure 2 repeats the information presented on figure 1. I suggest choosing only one of them (rather fig 2, as boxplot contains a lot of descriptive statistics).

2. On line 163 what did you mean by 'average'? Mean or median (as was stated on line 139)?

Author Response

Thank you for your revision. These are our replies to your comments:

1. Figure 2 repeats the information presented on figure 1. I suggest choosing only one of them (rather fig 2, as boxplot contains a lot of descriptive statistics).

- We removed figure 1, choosing fig. 2 with boxplot, as you suggested

2. On line 163 what did you mean by 'average'? Mean or median (as was stated on line 139)?

- We were considering the median. We changed the text according to

Reviewer 2 Report

Comments and Suggestions for Authors

This brief report manuscript explores the immune response and breakthrough infections among individuals with haemoglobinopathies. The data is interesting, addressing a gap in the literature. However, the methodology and presentation of immunological assessments need significant clarification and revision to enhance the quality and reliability of the findings. Below are my comments and suggestions for improvement.

Comments.

1. Figures 1 and 2: The two figures appear to represent the same dataset but use different visualisation formats.

To streamline the presentation, I recommend combining the figures. Use the individual scatter plots from Figure 1 and add error bars to display central tendency (e.g., median with IQR).

Box-and-whisker plot whiskers, which depict extremes, are unnecessary if individual data points are already displayed.

Refer to this example for guidance: https://www.graphpad.com/support/faq/plotting-the-geometric-mean-with-geometric-sd-error-bars

2. Terminology (IgG II Quant): This assay targets IgG anti-RBD, which is more specific than "anti-S." Use the term "anti-RBD" throughout the manuscript for precision.

References: https://www.fda.gov/media/146372/download

https://www.corelaboratory.abbott/int/en/offerings/segments/infectious-disease/sars-cov-2.html

3. Ceiling effect: Some samples exceed the assay’s calibration curve (>40,000 AU/mL or >5,680 BAU/mL).

If leftover samples are available, dilute these using the appropriate diluent (e.g., PBS or the instrument's specific diluent), reassess, and adjust the final concentration accordingly.

If re-measurement is not feasible, mention this limitation in the manuscript, as it may affect the accuracy of central tendency and error metrics.

4. Vaccine's name (Lines 91-92): The phrase "mRNA Comirnaty, Pfizer–BioNTech COVID-19 vaccine" is redundant since the vaccine type and manufacturer have already been mentioned earlier (Line 81). Use "Comirnaty" alone for brevity and clarity.

5. Serum Transferrin data: If data on serum transferrin levels were collected. I recommend including it in Table 1. This would provide additional valuable context regarding the baseline characteristics of the participants.

6. Misleading Statement (Lines 129-131, Figure 1): The statement "A measurement finding above 7 binding antibody units (BAU) was considered correlated with protection against the COVID-19 virus, according to the literature data reported [6,7]" is potentially misleading.

The antibody levels >7.1 BAU/mL (or 50.0 AU/mL) may indicate seropositivity but do not guarantee protection against infection. Vaccinated, convalescent or hybrid immunity individuals can still experience infections. 

Clarify that 7.1 BAU/mL is the cutoff for seropositive interpretation, not definitive protection.

7. Terminology (Lines 94-95): Replace "prior to vaccination" with "before vaccination" for better readability.

8. IgG Anti-N outcomes: If IgG anti-N levels were assessed, consider including this data in the manuscript. You can present the results as a visualisation (e.g., median with IQR) and compare groups, or provide a concise summary in the text if raw data is already in the supplementary materials.

Author Response

Thank you for your revision and suggestions. Attached below there are our replies.

  1. Figures 1 and 2: The two figures appear to represent the same dataset but use different visualisation formats.

To streamline the presentation, I recommend combining the figures. Use the individual scatter plots from Figure 1 and add error bars to display central tendency (e.g., median with IQR).

Box-and-whisker plot whiskers, which depict extremes, are unnecessary if individual data points are already displayed.

Refer to this example for guidance: https://www.graphpad.com/support/faq/plotting-the-geometric-mean-with-geometric-sd-error-bars

- We agree. Figure 1 was unnecessary and now there is one figure to describe the BAU values

  1. Terminology (IgG II Quant): This assay targets IgG anti-RBD, which is more specific than "anti-S." Use the term "anti-RBD" throughout the manuscript for precision.

References: https://www.fda.gov/media/146372/download

https://www.corelaboratory.abbott/int/en/offerings/segments/infectious-disease/sars-cov-2.html

- We changed the terminology in the manuscript, mentioning the antibodies as anti-RBD. We added the suggested references

  1. Ceiling effect: Some samples exceed the assay’s calibration curve (>40,000 AU/mL or >5,680 BAU/mL).

If leftover samples are available, dilute these using the appropriate diluent (e.g., PBS or the instrument's specific diluent), reassess, and adjust the final concentration accordingly.

If re-measurement is not feasible, mention this limitation in the manuscript, as it may affect the accuracy of central tendency and error metrics.

 - This is an interesting suggestion. Unfortunately, the sera are unavailable due to the long time from collection, making the re-measurement impossible. As you suggested, we added this concept in the discussion, underlying the limitation.

  1. Vaccine's name (Lines 91-92): The phrase "mRNA Comirnaty, Pfizer–BioNTech COVID-19 vaccine" is redundant since the vaccine type and manufacturer have already been mentioned earlier (Line 81). Use "Comirnaty" alone for brevity and clarity.

- We modified the text

  1. Serum Transferrin data: If data on serum transferrin levels were collected. I recommend including it in Table 1. This would provide additional valuable context regarding the baseline characteristics of the participants.

- Thank you for the suggestion. We add the transferrin values, being available and now collected

  1. Misleading Statement (Lines 129-131, Figure 1): The statement "A measurement finding above 7 binding antibody units (BAU) was considered correlated with protection against the COVID-19 virus, according to the literature data reported [6,7]" is potentially misleading.

The antibody levels >7.1 BAU/mL (or 50.0 AU/mL) may indicate seropositivity but do not guarantee protection against infection. Vaccinated, convalescent or hybrid immunity individuals can still experience infections. 

Clarify that 7.1 BAU/mL is the cutoff for seropositive interpretation, not definitive protection.

- Right observation. We modified the text following your suggestion

  1. Terminology (Lines 94-95): Replace "prior to vaccination" with "before vaccination" for better readability.

- We modified the text

  1. IgG Anti-N outcomes: If IgG anti-N levels were assessed, consider including this data in the manuscript. You can present the results as a visualisation (e.g., median with IQR) and compare groups, or provide a concise summary in the text if raw data is already in the supplementary materials.

- We mentioned the median and the range in the text. The data are reported in Supplementary File and we mentioned it in the text

Reviewer 3 Report

Comments and Suggestions for Authors

Overview:  Immune system impairment is frequently reported in patients affected by hemoglobinopathies due, in part, to iron accumulation, repeated antigenic stimulation from transfused blood products, chronic hemolysis, and a general hyperinflammatory state.  The authors evaluated anti-spike IgG production after 2 doses of vaccine for SARS-CoV-2 in patients affected by hemoglobinopathies.  All 114 enrolled patients (100%) developed adequate antibody production.  The amount of antibody was unrelated to any other clinical characteristics evaluated, including transfusion dependence or non-transfusion dependence, age, gender, disease type, ferritin, blood count, spleen status, and therapy with hydroxyurea or iron chelators.  47 (41.2%) of the patients developed break-through SARS-CoV-2 infection during the first 2 years of follow-up after vaccination, all with a mildly symptomatic course, without requiring hospitalization or experiencing a significative drop in hemoglobin values, allowing for a slight delay in their transfusion regimens.

Comments.

1.  Although the title references hemoglobinopathies in general, truth-in-advertising would suggest that the authors replace hemoglobinopathy in the title with thalassemia, specifically beta-thalassemia.  Only two patients had sickle cell disease.  The text should be edited appropriately.

2.  Given that this paper is from Italy, the lack of SCD is understandable.  The authors might wish to comment on ethnic origin, as well as age and sex.

3.  Mention is made of spleen status, given the significance of splenectomy in terms of subsequent risk for infection.  However, none of the figures or the table addressed spleen status.  This appears to be a significant oversight by the authors.  

4.  The authors rightly acknowledge the absence of a control group.  However, reference to reports of the likelihood of infection following vaccination would be helpful as an imperfect control. 

Author Response

Thank you for your revision and suggestions. Attached below there are our replies.

  1. Although the title references hemoglobinopathies in general, truth-in-advertising would suggest that the authors replace hemoglobinopathy in the title with thalassemia, specifically beta-thalassemia.  Only two patients had sickle cell disease.  The text should be edited appropriately.

- We think that mentioning only the beta-thalassemia in the title could be misleading and reductive, considering that there are 10 patients with Thalasso-drepanocytosis and 2 with sickle cell anemia. We agree with you that the text should be edited mentioning that the majority of patients were affected by beta-thalassemia, and we modified the text evidencing this concept

  1. Given that this paper is from Italy, the lack of SCD is understandable.  The authors might wish to comment on ethnic origin, as well as age and sex.

- Thank you for this suggestion. We include this discussion in the manuscript

  1. Mention is made of spleen status, given the significance of splenectomy in terms of subsequent risk for infection.  However, none of the figures or the table addressed spleen status.  This appears to be a significant oversight by the authors.

- Thank you for this suggestion. We add to table 1 the spleen status before the vaccination and discuss this concept in the manuscript

  1. The authors rightly acknowledge the absence of a control group.  However, reference to reports of the likelihood of infection following vaccination would be helpful as an imperfect control.

- Thank you for this suggestion. We discussed our data comparing them to the recent study published by Ghoti et al (Lancet, 2023)

Round 2

Reviewer 3 Report

Comments and Suggestions for Authors

All of my concerns have been appropriately addressed.

Author Response

Thank you for your time. We are happy to have addressed your interesting suggestions and improved the paper